# Tunable anisotropic van der Waals films of 2M-WS₂ for plasmon canalization

Qiaoxia Xing[1,9], Jiasheng Zhang[1,9], Yuqiang Fang[2,3,9], Chaoyu Song[1,9], Tuoyu Zhao[4,5], Yanlin Mou[1], Chong Wang[6,7], Junwei Ma[1], Yuangang Xie[1], Shenyang Huang [1], Lei Mu[1], Yuchen Lei[1], Wu Shi[4,5], Fuqiang Huang [2,3,8] ✉ & Hugen Yan [1] ✉

In-plane anisotropic van der Waals materials have emerged as a natural platform for anisotropic polaritons. Extreme anisotropic polaritons with in-situ broadband tunability are of great significance for on-chip photonics, yet their application remains challenging. In this work, we experimentally characterize through Fourier transform infrared spectroscopy measurements a van der Waals plasmonic material, 2M-WS₂, capable of supporting intrinsic room-temperature in-plane anisotropic plasmons in the far and mid-infrared regimes. In contrast to the recently revealed natural hyperbolic plasmons in other anisotropic materials, 2M-WS₂ supports canalized plasmons with flat isofrequency contours in the frequency range of ∼ 3000-5000 cm⁻¹. Furthermore, the anisotropic plasmons and the corresponding isofrequency contours can be reversibly tuned via in-situ ion-intercalation. The tunable anisotropic and canalization plasmons may open up further application perspectives in the field of uniaxial plasmonics, such as serving as active components in directional sensing, radiation manipulation, and polarization-dependent optical modulators.

Anisotropic metasurfaces are able to control light with more degrees of freedom than isotropic ones. Governed by the relationship of the optical conductivity components along two principal axes (imaginary part $\sigma''_{xx}$, $\sigma''_{yy}$, and real part $\sigma'_{xx}$, $\sigma'_{yy}$), elliptic ($\sigma''_{xx} \neq \sigma''_{yy} > 0$), canalization ($|\sigma''_{xx}| \gg |\sigma''_{yy}|$, or $\sigma''_{xx} \to \infty$ for energy ideally canalized along $x$-axis), and hyperbolic ($\sigma''_{xx}\sigma''_{yy} < 0$) momentum-space dispersions can be realized[1-6]. The variety of dispersion line shapes implies exotic electromagnetic phenomena, such as negative refraction, omnidirectional

energy collimation effect, and enhancement of the Purcell factor[6-11]. In recent years, natural in-plane anisotropic van der Waals materials have been proposed as host media to manipulate light with lower volumetric losses and are free of nanofabrication-induced challenges and damages compared to artificial metasurfaces[3,12]. Based on the in-plane anisotropy of phonons in van der Waals material α-MoO₃, the control of the phonon polariton dispersion at specific frequencies has been realized, mainly through twisting α-MoO₃ flakes or stacking α-MoO₃

[1]State Key Laboratory of Surface Physics, Key Laboratory of Micro and Nano-Photonic Structures (Ministry of Education), Shanghai Key Laboratory of Metasurfaces for Light Manipulation, and Department of Physics, Fudan University, 200433 Shanghai, China. [2]State Key Laboratory of High Performance Ceramics and Superfine Microstructure, Shanghai Institute of Ceramics, Chinese Academy of Sciences, 200050 Shanghai, China. [3]School of Materials Science and Engineering, Shanghai Jiao Tong University, 200240 Shanghai, China. [4]State Key Laboratory of Surface Physics and Institute for Nanoelectronic Devices and Quantum Computing, Fudan University, 200433 Shanghai, China. [5]Zhangjiang Fudan International Innovation Center, Fudan University, 201210 Shanghai, China. [6]Centre for Quantum Physics, Key Laboratory of Advanced Optoelectronic Quantum Architecture and Measurement (MOE), School of Physics, Beijing Institute of Technology, 100081 Beijing, China. [7]Beijing Key Lab of Nanophotonics & Ultrafine Optoelectronic Systems, School of Physics, Beijing Institute of Technology, 100081 Beijing, China. [8]Beijing National Laboratory for Molecular Sciences and State Key Laboratory of Rare Earth Materials Chemistry and Applications, College of Chemistry and Molecular Engineering, Peking University, 100871 Beijing, China. [9]These authors contributed equally: Qiaoxia Xing, Jiasheng Zhang, Yuqiang Fang, Chaoyu Song. ✉e-mail: huangfq@mail.sic.ac.cn; hgyan@fudan.edu.cn

with graphene[13–22]. With the conductivity ratio reaching a maximum at the transition from the elliptic to the hyperbolic topologies or vice versa, phonon polariton canalization with flat isofrequency contours facilitates near-field energy unidirectional propagation[23,24], which is desirable for diffractionless light guidance at the nanoscale using planar photonic devices[25,26]. However, the inherent narrow bandwidth and the lack of tunability associated with lattice vibrations limit the applications of canalized phonon polaritons.

The plasmon in metallic systems is a type of quasiparticle of collective charge oscillations, which has a variety of merits, such as enhancing light-mater interactions, offering a broad bandwidth, and being sensitive to multiple tuning methods especially for two-dimensional plasmons[27,28]. Therefore, natural in-plane anisotropic plasmons with high in-situ tunability are strongly desirable in low-symmetry photonics applications[2–5,29–51]. Natural in-plane hyperbolic plasmons have been experimentally demonstrated in WTe₂ and self-assembled aligned carbon nanotube films in the far- and mid-infrared regimes[52–54], respectively. Moreover, similar to the canalization of phonon polaritons, plasmonic materials with a large contrast in the optical conductivity between the principal axes will also cause the plasmons to preferentially propagate along directions with higher conductivity[2,4,5,8,30,34,38,39]. Yet, due to the rather swift transition from the elliptic to the hyperbolic regimes, the broadband canalization of plasmons in WTe₂ or other natural anisotropic van der Waals materials has not been directly demonstrated in experiments. Meanwhile, electrical tuning of plasmons in aforementioned films remains challenging.

Van der Waals semimetal 2M-WS₂ has attracted extensive attention due to its coexistence of topological surface states and intrinsic superconductivity with the highest transition temperature among transition metal dichalcogenides[55]. Furthermore, the in-plane anisotropy and Li⁺ intercalation tunable electrical transport properties[55,56], facile mechanical exfoliation, and high mobility (~7000 cm² V⁻¹ s⁻¹ at 10 K[55]) render 2M-WS₂ a promising candidate for tunable natural in-plane anisotropic plasmonic materials.

Here we experimentally demonstrate 2M-WS₂ as an infrared anisotropic material through Fourier transform infrared spectroscopy. Moreover, anisotropic plasmons from far- to mid-infrared regimes are examined in patterned 2M-WS₂. Above the frequency of 3000 cm⁻¹, the plasmons along one direction diminish, while they persist along another direction. This behavior leads to the channel-like isofrequency contours in momentum space over a relatively broad frequency range from 3000 cm⁻¹ to 5000 cm⁻¹, suggesting broadband plasmon canalization. More importantly, ion-intercalation renders anisotropic plasmons and the corresponding isofrequency contours reversibly tunable. We also corroborate the experimental results with finite element simulations.

## Results

### In-plane anisotropic infrared absorption in 2M-WS₂ thin films
The metallic-phase 2M-WS₂ is a layered semimetal with in-plane orthogonal principal axes, belonging to the monoclinic crystallographic system. The crystal structure is shown in Fig. 1a. Polarized Raman spectra in Supplementary Fig. 1 reveal the anisotropy of 2M-WS₂ crystal. Exfoliated 2M-WS₂ was characterized through Fourier transform infrared spectroscopy, as illustrated in Fig. 1b. The extinction spectra $1-T/T_0$ were recorded, where $T$ and $T_0$ represent the transmission of the infrared light through the sample and the bare substrate, respectively. The synthesized 2M-WS₂ crystal was exfoliated into films with a thickness of ~50 nm. Figure 1c shows a typical optical image of a sample on a diamond substrate, with the principal axes denoted as "b" and "c". The film area is ~50 × 100 μm². Infrared extinction spectra measured at room and liquid nitrogen temperatures with the light polarized along the b- and c-axis are shown in Fig. 1d. The spectra exhibit significant polarization dependence, while the temperature dependence is modest. Lowering the temperature makes

features in the absorption spectra slightly sharper. The measured extinction spectra were fitted using the optical conductivities of Drude response and interband absorptions, as shown in Supplementary Fig. 2 (see the details in "Methods" section and the Supplementary Note 1). The fitted Drude weights along the b- and c-axis at room temperature are $2.9 \times 10^{13}\ \Omega^{-1}\ s^{-1}$ and $1.1 \times 10^{13}\ \Omega^{-1}\ s^{-1}$, respectively. Since the Drude weight $D$ for the material with a parabolic band structure is governed by $D = \pi n e^2 / m^*$, where $n$ and $m^*$ are the carrier density and effective mass of the material, respectively, $e$ is the elementary charge. The effective mass ratio for the two principal axes $m_c^*/m_b^*$ is ~2.6 at room temperature. The fitted imaginary (real) part of the conductivity ratio between the b-axis and c-axis is ~40 at 6000 cm⁻¹ (with a maximum of ~4 in the measurement range), as illustrated in the Supplementary Fig. 2. Compared to the theoretically proposed plasmon canalization in black phosphorus with the anisotropic imaginary part of conductivity ratio of ~3–15[4,30,38], the large anisotropy of the 2M-WS₂ film potentially supports extreme anisotropic (canalized) plasmons.

### Anisotropic infrared plasmons in 2M-WS₂ disk arrays
After the in-plane anisotropy in thin films of 2M-WS₂ was characterized, the exfoliated films were patterned into disk arrays with disk diameter of ~8 to 0.3 μm. This geometrically isotropic structure enables us to characterize the plasmons by far-field spectroscopy and reveal the anisotropy of plasmons resulted from the intrinsic properties of the material. As shown in Fig. 2a, with decreasing disk diameter, plasmons along the b-axis exhibit a monotonic blueshift from ~300 cm⁻¹ to 6300 cm⁻¹, while when the light polarization is switched to the c-axis, the plasmons also blueshift but gradually disappear ~3000 cm⁻¹. The resonance frequency differences of the plasmons along the two axes are not so prominent in the far-infrared regime (e.g., the 8 and 6 μm disks in Fig. 2a), although both the plasmon strength and fitted Drude weight of the film reveal apparent differences. This scenario can be explained by invoking the retardation effect. In specific, when the 2D conductivity of a film is comparable with the velocity of light (they have the same units in the Gaussian units)[57,58], field dynamics should be taken into account, and plasmon dispersions are primarily governed by the geometry of the nanostructures and the dielectric environments[59], rather than the intrinsic optical conductivity of the 2D material. Figure 2b shows the extinction spectra of the same samples excluding the two largest ones (disk diameter of 8 and 6 μm) at 81 K. The inset is a scanning electron microscope (SEM) image of a typical disk array. Decreasing temperature results in a reduction of the plasmon linewidth, especially for polarization along the c-axis, but the change in the spectra weight is insignificant. Figure 2c displays the finite element simulated extinction spectra of a disk suspended in air, based on the fitting parameters of the room temperature unpatterned film in Fig. 1. The simulated frequency ratio of plasmons along b- and c-axis is about 1.36, which is consistent with the corresponding experimental value of ~1.27 (see Supplementary Note 2 for the detailed information). For further analysis, we fit the measured extinction spectra as shown in Fig. 2d (see "Methods" section and the Supplementary Note 1 for detailed information). Although more than one interband transition and probably higher-order plasmon modes exist, we use one Lorentz peak for the plasmon main peak and two Lorentz peaks for others to fit the spectra due to the lack of prominent interband transition features[28]. The fitted spectra agree well with the experimental measurements.

### Plasmon dispersion and isofrequency contour in 2M-WS₂
The extracted plasmon parameters from the experimental data in Fig. 2a, b are analyzed further in Fig. 3. Figure 3a shows the plasmon dispersion along the b- and c-axis (right and left panels) at room temperature and 81 K (red and blue spheres). For patterned disks with a diameter $d$, the equivalent magnitude of wave vector $\mathbf{q}$ is $3\pi/4d$[60]. With the increase of the wave vector $\mathbf{q}$, the resonance frequencies

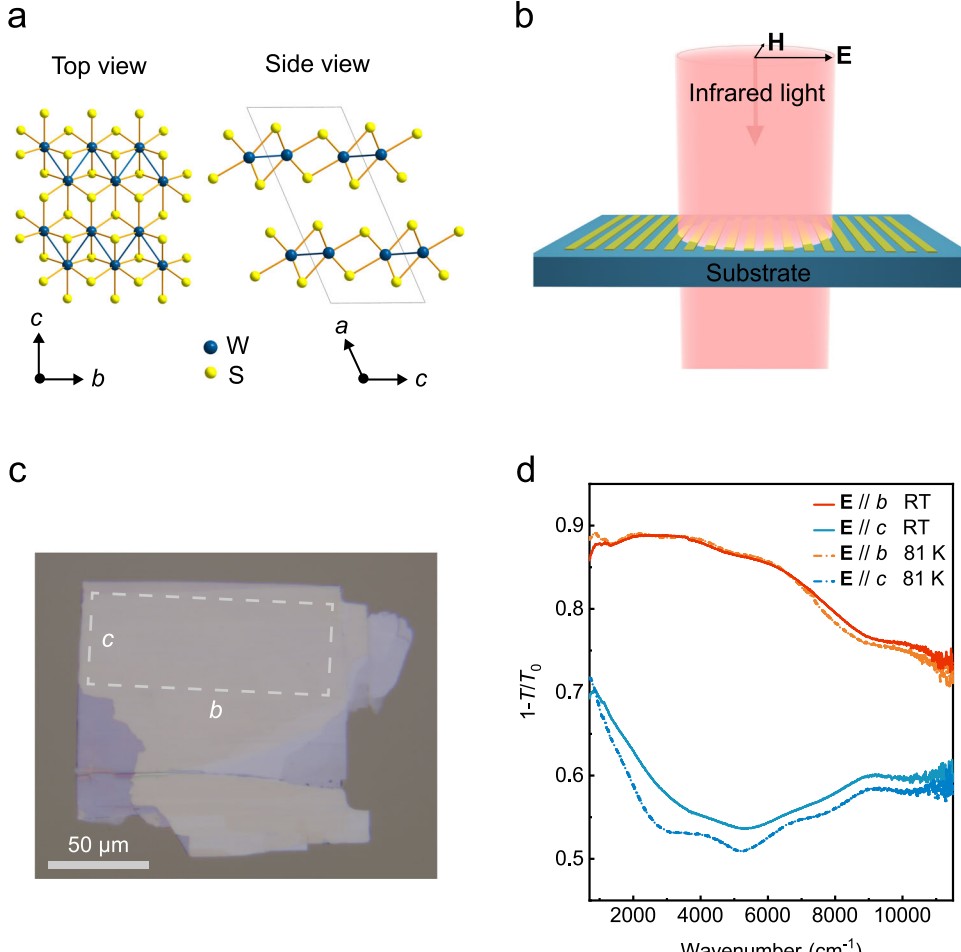

**Fig. 1 | Characterization of a typical 2M-WS₂ film. a** An illustration of the 2M-WS₂ crystal structure, with the unit cell marked by the light-gray frame. **b** A scheme for the infrared spectra measurement with polarized light, where the in-plane electric field **E** and magnetic field **H** are indicated. **c** An optical image of a typical 2M-WS₂ film with a thickness of ~50 nm on a diamond substrate. The dashed box in the image represents the measurement area, and the principal axes of the film are indicated as "*b*" and "*c*". The scale bar is 50 μm. **d** The extinction spectra with light polarization along the two principal axes of the 2M-WS₂ film in **c** at room temperature and 81 K.

---

along both the *b*- and *c*-axis increase, and the frequency difference between them becomes larger. In contrast to the $\omega \propto \sqrt{q}$ dispersion along the *c*-axis, it exhibits an almost linear dispersion in the frequency range of ~300–6300 cm⁻¹ along the *b*-axis. The anisotropic dispersion can also be verified through the loss function -Im(1/ε) with the retardation effect taken into account, where ε is the RPA dielectric function of the material (see Supplementary Note 3 for the calculation of the loss function). The conductivity in the 2D case used in the loss function was extracted from the extinction spectra of the unpatterned film in Fig. 1d. The calculated loss function is basically consistent with the experimentally obtained plasmon dispersion. Neglecting the conductivity related to interband transitions with frequencies beyond our measurement range mainly contributes to the deviation in the plasmon strength[61]. In Fig. 3b, the fitted linewidths of the plasmons from spectra in Fig. 2 are displayed. Along the *b*-axis, the plasmon linewidth linearly increases from 170 cm⁻¹ to 3400 cm⁻¹ with the increase of the plasmon frequency. This behavior is a result of the combined effects of the retardation (lower frequency range) and interband transitions (higher frequency range). The linewidth of plasmons in disks scales linearly to **q** when accounting for retardation[59]. Given the linear dispersion along the *b*-axis and the aforementioned relation $\Gamma_p \propto q$, the linear fit in Fig. 3b (light gray dashed line) is well justified. While the plasmon dispersion along the *c*-axis follows $\sqrt{q}$ scaling, and the plasmon linewidth correlates positively with the corresponding wave

vector, the relation $\Gamma_p \propto \omega^2$ (light gray dashed dot line) can roughly capture the plasmon frequency-dependent linewidth along the *c*-axis. And above the frequency of ~3000 cm⁻¹, the rapidly increasing linewidth wipes out the plasmon feature in the spectra. Moreover, the coupling between the plasmon and interband transitions along the *c*-axis at a frequency of beyond 3000 cm⁻¹ also leads to diminishing plasmon oscillator strength (see Supplementary Note 4 for the evolution of the plasmon spectrum weight). The propagation characteristics of the plasmons can be manifested by the corresponding isofrequency contour. Based on the extracted conductivity of the film, the isofrequency contours can be predicted via the maximum of the loss function (see the details in the Supplementary Note 5), as shown in the Fig. 3c. The axis ratio $q_c/q_b$ of the elliptic isofrequency contour increases from ~2.3 at 1000 cm⁻¹ to ~3.5 at 3000 cm⁻¹. Eventually, the gradually elongated ellipse changes to flat isofrequency contours as the frequency further increases.

To experimentally determine the plasmon isofrequency contours of the 2M-WS₂ in the two-dimensional momentum space through far-field spectroscopy, it's necessary to obtain plasmon frequencies with wave vector **q** along various directions as well. Microstructure arrays in the ribbon form are specifically suitable to meet this purpose, rather than disks with plasmons only along the principal axes measurable in experiment. Therefore, we patterned the films into a set of skew ribbons with different widths and skew angles between the perpendicular

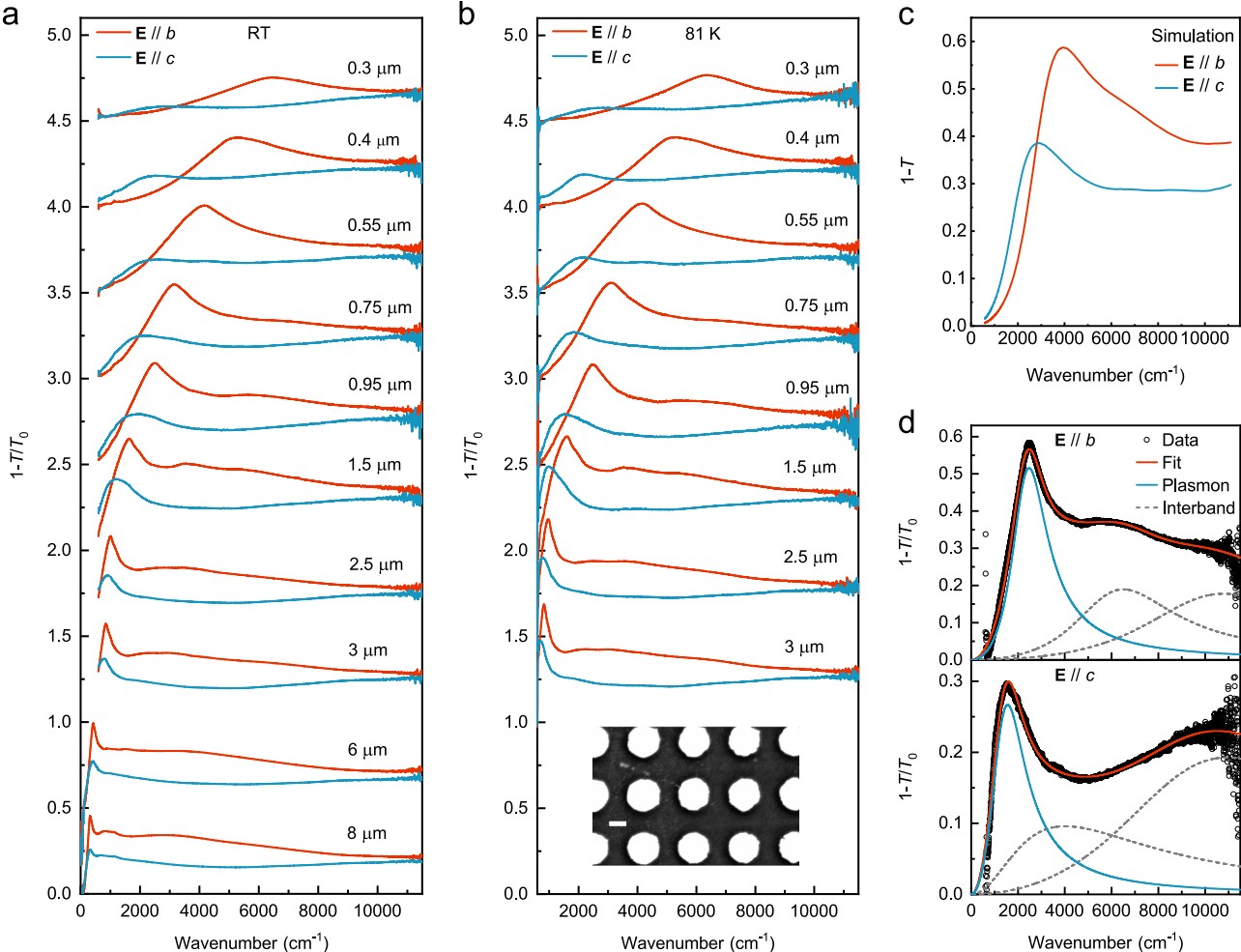

**Fig. 2 | Plasmons in 2M-WS₂ disk arrays. a** Plasmons along the two principal axes of the 2M-WS₂ disks with diameters from 8 to 0.3 μm at room temperature, and the spacing between the disks is 0.65 times their diameter. Spectra for different disk arrays are vertically offset for clarity. **b** Measured mid- to near-infrared spectra at 81 K of the same disks, except for the two largest ones with plasmon frequencies in the far-infrared regime, as the sample area is not sufficient for long-wavelength infrared measurements. The inset is an SEM image of a typical 2M-WS₂ disk array, and the scale bar is 400 nm. Spectra are vertically offset for clarity. **c** The extinction spectra of a suspended 2M-WS₂ disk with a diameter of 1 μm, simulated using the finite element method based on fitting parameters of the film in Fig. 1. **d** An example showing the plasmon spectra fitting results of the disk array on a diamond substrate with a diameter of 0.95 μm in **b**. The black symbols, red, blue and gray dashed lines represent experimental data, along with the corresponding fitting curves of the overall extinction spectra, Drude and interband transition components, respectively.

direction of the ribbons and the $b$-axis from 0° to −90°. The inset of Fig. 4a shows an SEM image of a typical skew ribbon array and the definition of the skew angle $\theta$, where the direction parallel to the $b$-axis is defined as 0°. The skew ribbons were measured with Fourier transform infrared spectroscopy and the resonance frequencies were determined (see Supplementary Fig. 7 for the extinction spectra of the skew ribbons). Consequently, the dispersions of the plasmons with wave vector $\mathbf{q}$ ($q = \pi/L$, $L$ is the ribbon width) along different directions are plotted, as shown in Fig. 4a. Additionally, the dispersions are fitted through the maxima of the loss functions (see Supplementary Note 5 for the detailed information). Since the dispersions of plasmons with wave vector $\mathbf{q}$ along different angles have been plotted and fitted, the in-plane isofrequency contours are reconstructed at frequencies of 2006 cm⁻¹, 2849 cm⁻¹, 3676 cm⁻¹, 4242 cm⁻¹, 4450 cm⁻¹, and 5085 cm⁻¹, as presented in Fig. 4b. The corresponding resonance frequencies are also indicated in Fig. 4a. The normal direction to the isofrequency contour is the plasmon propagation direction. Therefore, plasmons can propagate along all directions when the resonance frequencies are below 3000 cm⁻¹, since the isofrequency contours are elliptic. Above 3000 cm⁻¹, the contours are nearly straight parallel lines, suggesting

nearly unidirectional plasmon propagation in films or localized electric field in nanostructures along the $b$-axis. The isofrequency contours and corresponding near field electric field distribution |**E**| (see Supplementary Fig. 4) indicate the canalization capability.

## Reversibly tunable plasmons in 2M-WS₂

Two-dimensional materials provide a great platform for developing tunable photonic devices. To achieve dynamic tunability in photonic devices made of thin films, ion intercalation is more effective than regular electrostatic gating. Figure 5a depicts a schematic of gate-controlled ion intercalation through ion-gel. Figure 5b shows the extinction spectra of Li⁺ intercalated ribbons parallel to the $c$-axis, with the ribbon width of 0.6 μm and the electric field of the polarized infrared light along the $b$-axis. With increasing resistance, the plasmon redshifts and the strength decreases. Importantly, the tunability of the optical response and the resistance is reversable, as shown in Fig. 5c (see Supplementary Fig. 9 for more information). The highly tunable carrier density of 2M-WS₂ also leads to the modulation of the film absorption. As shown in Supplementary Fig. 8b, c a film with the source and drain along one of the principal axes is intercalated via Li⁺ ion-gel,

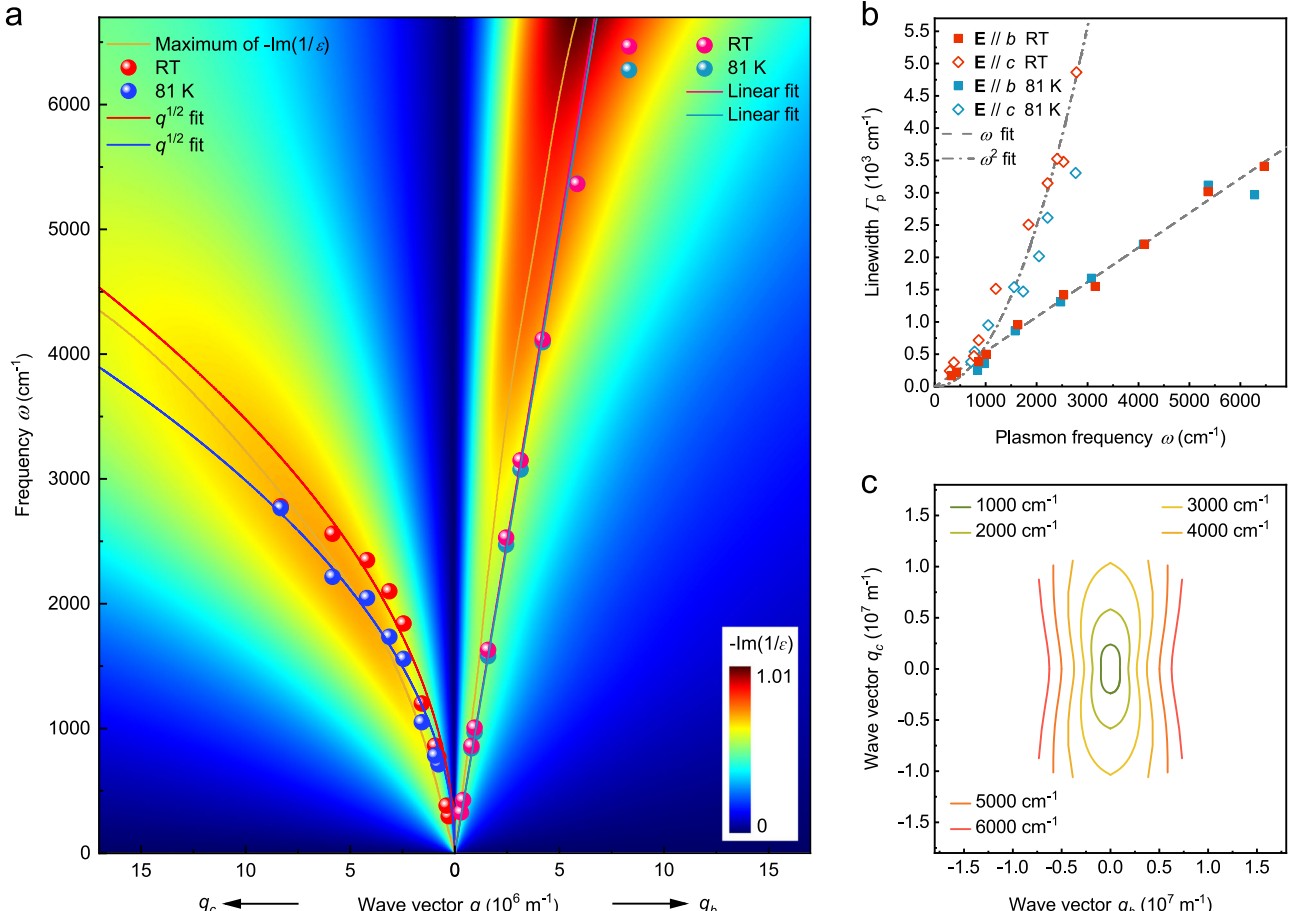

**Fig. 3 | Dispersions and linewidths of the plasmons in 2M-WS$_2$ disks.**
**a** Dispersions of plasmons along the $b$- and $c$-axis in 2M-WS$_2$ disks at room temperature and 81 K, represented by red and blue spheres, respectively. The relations between plasmon frequency $\omega$ and wave vector $q$, which scale as $\omega \propto \sqrt{q}$ and $\omega \propto q$ were used to fit the plasmon dispersions along the $c$- and $b$-axis, respectively. The loss function at room temperature is shown as a pseudocolor map. **b** Extracted linewidth versus frequency of the plasmons in Fig. 2a, b. The light gray dashed and dashed dot lines represent the corresponding fitting curves. **c** Isofrequency contours of the plasmons obtained through maxima of the loss functions and the extracted conductivities of the film in Fig. 1.

and the extinction spectra exhibit a reduction of the Drude response as the resistance of the film increases. Due to the large Li$^+$-induced damping along the $c$-axis which is manifested by the extinction spectra of the film along the $c$-axis, plasmon shifts cannot be detected before the plasmons fully diminish in strength.

Other media for the intercalation are also possible. We further tuned the carrier density of the 2M-WS$_2$ through H$^+$ intercalation from gate-controlled electrolyzed deionized water[62]. To avoid extra absorption, water droplets were blown off and samples were dried through nitrogen gas before measurements. As shown in Fig. 5d, e, the tunability of plasmons along the $b$- and $c$-axis is also demonstrated. The plasmons redshift and the strength decreases with increasing resistance. The redshift of the plasmon along the $b$-axis is ~17% of the original frequency (Fig. 5d), similar to the corresponding value of ~20% in Fig. 5b with Li$^+$ ion-gel. The plasmon along the $c$-axis redshifts from 2200 cm$^{-1}$ to a frequency below 800 cm$^{-1}$, which is the lower measurable limit with CaF$_2$ substrates. The amount of the frequency redshift is larger than 63% along the $c$-axis, manifesting significant tunability of ion intercalation. Figure 5f presents the resistance-dependent frequency of the plasmons.

The change of the plasmons with ion intercalation can be further confirmed through extinction spectra of the intercalated unpatterned 2M-WS$_2$ film. As shown in Supplementary Figs. 10 and 11, intercalation-dependent extinction spectra of a film and the corresponding fitting results are presented. Drude weights decrease with increasing ion intercalation, while the peak positions and the strength of the

interband transitions remain substantially fixed, simplifying the analysis of the intercalation effect on the plasmon. The extracted Drude weight along the $b$-axis decreases linearly with that of the $c$-axis, as shown in Fig. 5g. The slope is about 2.6, which is consistent with the ratio of $D_b/D_c = 2.6$ extracted from Fig. 1d. With the Drude weight decreasing to 0.1 (0.15) times of the original along the $c$-axis ($b$-axis) after intercalation, the corresponding loss function changes from Fig. 3a to Fig. 5h. A significant redshift of the plasmon along the $c$-axis with the value of ~88% is in agreement with the experimental result in Fig. 5e. More importantly, with the Drude weight decreasing to about one order smaller than the original value and the interbands almost intact, the original canalization frequency range shifts to lower frequency. We take the lower canalization frequency range limit of 3000 cm$^{-1}$ as an example to demonstrate the change of the calculated isofrequency contours. As shown in Fig. 5i, the elliptic isofrequency contour gradually enlarges and eventually changes to a flat shape as the intercalation progresses (see Supplementary Note 6 for detailed information). The extinction spectra after deintercalation and additional spectra of ion intercalated 2M-WS$_2$ films ensuring the reproducibility of the tuning are displayed in Supplementary Figs. 9 and 13.

## Discussion

In summary, van der Waals semimetal 2M-WS$_2$ has been experimentally investigated as an in-plane anisotropic plasmonic material. Anisotropic plasmons ranging from far- to mid-infrared regime have been demonstrated at room and liquid nitrogen temperatures via Fourier

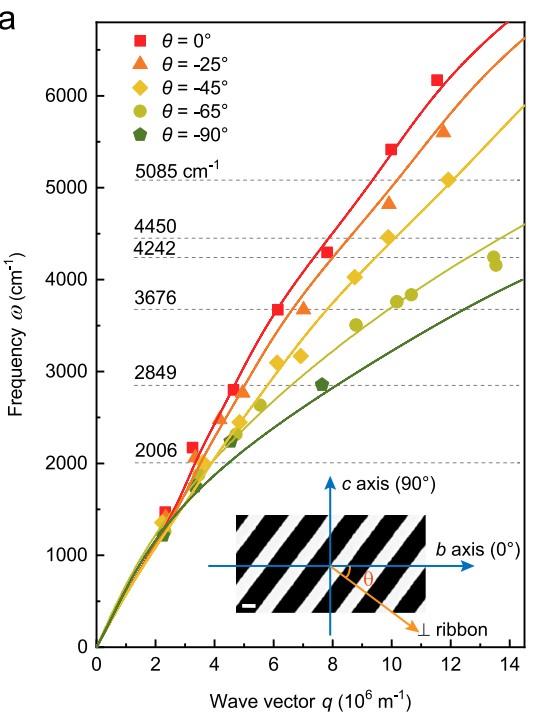

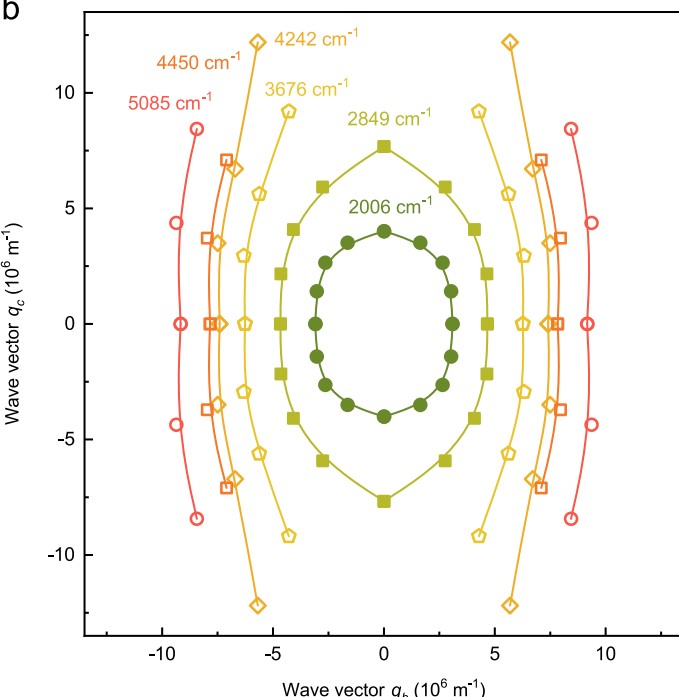

**Fig. 4 | Plasmon dispersions and isofrequency contours in 2M-WS₂ in 2D wave vector space. a** Dispersions of the 2M-WS₂ ribbons at room temperature with different skew angles $\theta$, as shown in the figure legend. The corresponding frequency cutting lines are indicated by the light gray dashed lines. The inset is an SEM image of a skew ribbon array, and the definition of the skew angle $\theta$ is indicated. The scale bar is 650 μm. **b** Isofrequency contours of 2M-WS₂ plasmons plotted according to the frequency cutting lines as indicated in **a**. B-spline curves are guides to the eye.

transform infrared spectroscopy measurements of the patterned nanostructures. The nearly linear and $\omega \propto \sqrt{q}$ plasmon dispersions along the $b$- and $c$-axis are determined, respectively. The isofrequency contours and near-field electric field distributions demonstrate elliptic plasmons below 3000 cm⁻¹ and a broadband plasmon canalization regime at ~3000–5000 cm⁻¹. Furthermore, reversibly tunable spectral responses of the plasmons and the corresponding isofrequency contours via in situ ion-intercalation are demonstrated. Our study paves the way for exploring the electrically switchable unidirectional plasmonic devices.

## Methods

### Sample preparation

Van der Waals semimetal 2M-WS₂ single crystals were grown via oxidizing the precursor K₀.₇WS₂ crystals. To synthesize K₀.₇WS₂, K₂S₂ (prepared via liquid ammonia), W (99.9%, Alfa Aesar), and S (99.99%, Alfa Aesar) were mixed in stoichiometric ratios and grounded in an argon-filled glovebox. The resulting mixtures were pressed into a pellet and sealed in an evacuated quartz tube. The tube underwent heating to 850 °C for 2000 minutes, followed by gradual cooling to 550 °C at a rate of 0.1 °C min⁻¹. The synthesized K₀.₇WS₂ (0.1 g) was chemically oxidized using K₂Cr₂O₇ (0.01 mol L⁻¹) in aqueous H₂SO₄ (50 mL, 0.02 mol L⁻¹) at room temperature for 1 h. Ultimately, the 2M-WS₂ crystals were obtained after multiple washes with distilled water and drying in a vacuum oven at room temperature. The prepared crystals were mechanically exfoliated into films on diamond substrates (CaF₂ substrates for ion intercalation experiments). The film thickness was determined by a stylus profiler (Bruker DektakXT) and the mid-infrared transmission spectra. Subsequently, through the standard electron beam lithography and reactive ion etching via carbon tetra-fluoride (CF₄) reactive gas, the films were patterned into nanostructures such as nano-disks and ribbons. Note that the electron beam resist PMMA was baked at a relatively low temperature of 85°C for

5 minutes due to the metastable phase of 2M-WS₂, which cannot remain stable at the usual baking temperature of 180 °C. For Li⁺ intercalation, Au/Cr contacts were deposited onto stencil mask coated samples using thermal evaporation. A mixture of LiClO₄ and polyethylene oxide was used as a solid electrolyte for in-situ intercalation and the corresponding measurement. As an alternative intercalation method, the samples can be tuned through deionized water with the resistance of ~1–2 MΩ. The gate voltage was kept at 1 V and the resistance of the sample was tuned by controlling the time of applied voltage. The transmission spectra were measured after drying the water with nitrogen gas.

### Infrared spectra measurement

Bruker Fourier transform infrared spectrometer (Vertex 70 v) integrated with a Hyperion 2000 microscope along with the far- and mid-infrared polarizers were used in the measurement. For mid to near-infrared and far-infrared spectra, the MCT (Mercury Cadmium Telluride) and liquid-helium-cooled silicon bolometer (IR Labs) were employed as the detectors, respectively. Low-temperature measurements were conducted using a cooling and heating chamber (Linkam, FTIR600). The ion intercalation procedure was performed using a lock-in amplifer (Stanford Research 830) and a Keithley 2612B sourcemeter.

### Spectra fitting procedures

The extinction spectrum $1 \text{-} T/T_0$ is governed by the complex optical conductivity $\sigma(\omega)$:

$$1 - \frac{T}{T_0} = 1 - \frac{1}{|1 + Z_0 \sigma(\omega)/(1+n_s)|^2} \qquad (1)$$

where $Z_0$ is vacuum impedance, and $n_s$ is the refraction index of the substrate. Intraband and interband transitions contribute to the

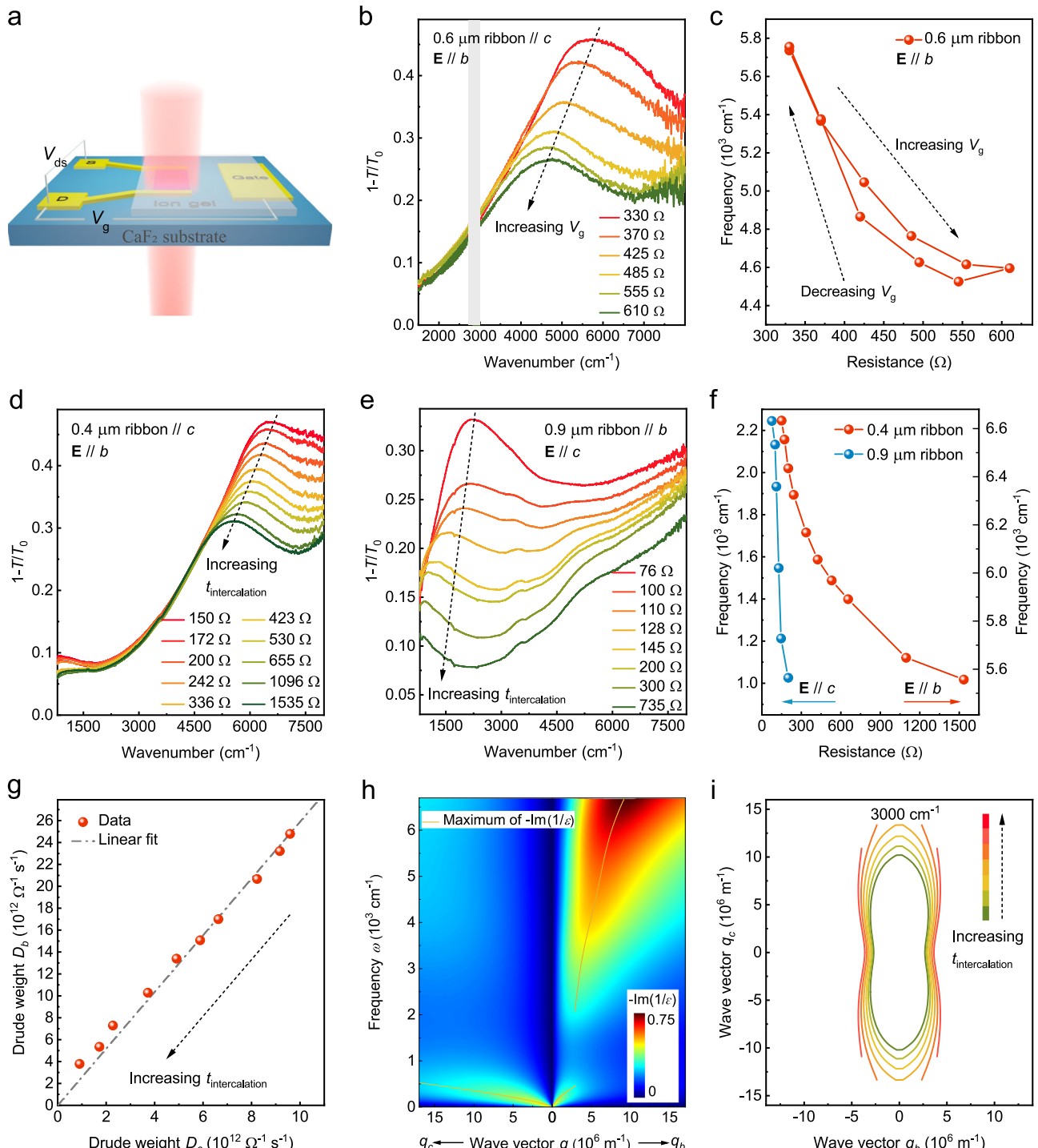

**Fig. 5 | Tunable plasmons in 2M-WS₂ through ion-intercalation. a** An illustration of the infrared spectrum measurement of an ion intercalated sample. $V_g$ represents the electrochemical intercalation voltage applied between the ion gel and the sample. The voltage between source and drain $V_{ds}$ serves to monitor the sample resistance. **b** The evolution of the plasmon spectrum in a 0.6 μm wide $c$-axis oriented ribbon array with different resistances under gate-controlled Li⁺ intercalation. Due to the strong absorption of the ion-gel below 1500 cm⁻¹ and the other region with shade of gray in the figure, these frequency regimes are not displayed. **c** Reversible resistance-dependent plasmon frequency extracted from **b**. The red line is a guide to the eye. **d**, **e** The evolution of the plasmons with increasing intercalation duration $t_{intercalation}$ in ribbon arrays along the $b$- and $c$-axis under gate-controlled deionized water, respectively. **f** Resistance-dependent plasmon frequencies extracted from **d** and **e**. The red and blue lines are guides to the eye. **g** Extracted Drude weights of a film along the $b$- and $c$-axis at different intercalation durations. They are proportional to each other. **h** Calculated loss function using the extracted film parameters after intercalation. **i** Calculated intercalation-dependent isofrequency contours at the frequency of 3000 cm⁻¹.

total film conductivity. For the spectra of patterned nanostructures, Lorentz oscillators are used to fit the optical conductivity of plasmons and interband transitions. See the Supplementary Note 1 for more details.

## Data availability

All relevant experimental data are presented in the paper and the Supplementary Information. The data that support the findings of this study are available from the corresponding authors upon request.

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

## Acknowledgements

H.Y. is grateful to the financial support from the National Key Research and Development Program of China (Grant Nos. 2022YFA1404700, 2021YFA1400100), the National Natural Science Foundation of China (Grant No. 12074085), the Natural Science Foundation of Shanghai (Grant Nos. 23XD1400200, 23JC1401100). Y.F. is grateful to the financial support from the National Natural Science Foundation of China (Grant No. 52103353) and Shanghai Rising-Star Program (Grant No. 23QA1410700). Q.X. is grateful for the financial support from the China Postdoctoral Science Foundation (Grant No. 2021M700861). W.S. is supported by the National Natural Science Foundation of China (Grant No. 12274090) and the Natural Science Foundation of Shanghai (Grant No. 22ZR1406300). C.W. is grateful for the financial support from the National Key Research and Development Program of China (Grant No. 2022YFA1403400) and the National Natural Science Foundation of China (Grant Nos. 12274030 and 11704075). The authors thank JiaJun Wang and professor Lei Shi for the help in measuring the visible spectra of the films, and professor Yaomin Dai for measurement of infrared spectra. The authors acknowledge Weiliang Ma for useful discussions on the finite element simulation. Part of the experimental work was carried out in the Fudan Nanofabrication Lab.

## Author contributions

H.Y., C.S. and Q.X. conceived the experiments. Y.F. and F.H. synthesized the 2M-WS$_2$ bulk crystals. Q.X. prepared the samples with help from J.Z. and Y.M.; Q.X. performed the measurements with assistance from J.Z., T.Z., C.W., J.M., Y.X., S.H., L.M. and Y.L.; T.Z. and W.S. prepared the ion gel. Q.X. performed the finite element simulation. Q.X. and H.Y. analyzed the data and wrote the paper, with inputs from F.H. and W.S.; H.Y. supervised the whole project. All authors commented on the paper.

## Competing interests

The authors declare no competing interests.
