## [Peer Review File · Nature Communications]

Tunable anisotropic van der Waals films of 2M-WS_2 for
plasmon canalizationREVIEWER COMMENTS

Reviewer #1 (Remarks to the Author):

The work presents a great experimental part that shows evidences of plasmonic anisotropy, however the points below need elucidation.

1 – The Raman analysis, used to show the crystalline anisotropy, is only superficially discussed. As anisotropy is an important feature for the work, one should further the explanation of the Raman spectra and clearly indicate, if possible, citing proper literature, which information confirms the anisotropy.

2 – In the line 146 of the “Anisotropic infrared plasmons in 2M-WS₂ disk arrays” section, the small difference between plasmons resonant frequencies of the two largest disks in the far-IR is explained by the retardation effect that is important when the 2D conductivity becomes comparable to the light velocity in Gaussian units. In this case, the plasmon dispersion is governed by the nanostructures’ geometry and the dielectric environment according to the presented discussion. This part rises some questions:

2.1 – Which are the 2D conductivity values for those disks that allow assuming the retardation effect? Are these values different from those obtained for the fits in Fig. 1 for the bare crystal?

2.2 – If the retardation effect is relevant for the 8 and 6 micrometer-sized diameter disks, why does it not cause a similar influence on the smaller disks of nanometer-sized diameters?

2.3 – As the dielectric environment must be considered, how was the substrate computed in the modelling?

2.4 – A more technical aspect: what is the spacing between disks in each patterning?

3 – In the fit modelling to the bare crystals, it was used 1 term for the Drude response of free charges and 6 terms accounting for interband transitions. However, to extract the plasmons parameters from the disk-patterned crystals for a same spectral range, the fit modelling considers 3 three terms: 1 for the plasmon and 2 for interband transitions. Could the authors explain the reason the amount of interband components is different in each case? Based on the interband transitions fundamentals that is determined by the material electronic configuration, is it expected to vary due to the patterning in comparison to the bare crystal?

4 – A question related to the previous, it is used the fit extracted conductivity from the unpattern crystal, which has the 6 interband components, as input to calculate the dispersion from the loss function. But the plasmon experimental dispersion is explained (fitted) by the modelling with 3 interband components. These approaches seem to have discrepancies.

5 – It is said that the loss function calculation considers the retardation effect, however there is no discussion on why it is needed. What would change in the dispersion if retardation was not taken into account?

6 – On the isofrequency contours of Fig. 4.

6.1 – Could the authors describe the fitting modelling for skew ribbons? Which were the Drude weights and how many interband components were used in each case? Is this approach consistent with the modelling of the disks’ cases?

6.2 – How was q determined? In the disks’ case it comes from the relation in line 187, but the one for

ribbons is not explicitly informed in the text.

6.3 – The theoretical dispersion curves in Fig. 4b are obtained by fitting the loss function maxima to the experimental data. This procedure corresponds to determining the material conductivity that is a material constant, only changeable under extreme states (high magnetic fields, temperature, and pressure, for instance), and was already evaluated from the fits in Fig. 3a. It is not clear such a change in the conductivity from the disk to the ribbon patterned WS₂. This point is very critical.

6.4 – Which was the spacing in between the ribbons for each angle? Could the authors comment on how the spacing would change the dipole coupling, which is discussed in supplementary information note 5, and that implied in the 0.43 scaling factor to adjust the isofrequencies contours in Fig. 4?

6.5 – The shown canalization in Fig. 4b is not reproduced by the theoretically predicted ones in supplementary Fig. 4. Could the reasons for that be presented?

Reviewer #2 (Remarks to the Author):

There has been substantial interest in creating a toolbox of materials which could be valuable in developing planar structures to control IR radiation for sensing and advanced photonics. 2D materials with natural anisotropies in the IR are ideal candidates in this space. The paper explores 2M-WS₂ for this application. This work in many ways simply builds on earlier work on WTe₂ by the same group. (Ref. Nat Commun 11, 1158 (2020) and Light Sci Appl 12, 193 (2023)). The study shows that this materials system is anisotropic and can support plasmon polaritons in the 2-3micron range (3000cm⁻¹ - 5000cm⁻¹). While these observations are noteworthy, they are not substantially different than earlier work. One interesting new feature is the ability to tune the system using electrically driven ion-intercalation. This strategy for active control is both novel and interesting. Several questions need to be addressed. First there is a substantial difference in the resonance peaks in figure 2 between the simulation figure 2C and figures 2A&B for ~1micron diameter structures. No explanation is offered. This needs to be addressed. The authors should focus more on the tunability aspect of this system, there are several interesting features in their data which they don't try to explain. For example, what drives the O~1000cm⁻¹ red shift observed in figure 5D? It's not as pronounced in figure 5E. This work is of interest to the community but the authors should focus on the truly novel parts of their study. Restructuring the paper with increased focus in the viability of ion-gel based tunability would make this a stronger submission.

I believe this paper would be worthy of publication with substantial revision.

Reviewer #3 (Remarks to the Author):

1. This work utilized far-field spectroscopy on nano-patterned devices to demonstrate an anisotropic plasmon in 2M-WS₂ thin films.
2. The authors systematically studied the isofrequency contours of the TMD material and demonstrated

an elliptical plasmon polariton dispersion with broadband canalization.

3. The authors further demonstrated the tunability of the anisotropic plasmonic response using ion gel gating.

4. Prior work mainly focuses on exciton-polariton in TMD materials and this work is the first to demonstrate tunable anisotropic plasmon polariton induced by the Drude and interband response.

5. Compared to prior works from their group, this work demonstrated the tuning of plasmon polariton through ion gel in the mid-infrared range. This work also demonstrated broadband canalization of plasmon that enables unidirectional propagation.

The authors could comment on whether there are other ways to tune the system, such as optically induced hyperbolicity [1].

This work should be published with minor revisions.

[1] Sternbach et al., Science 371, 617–620 (2021)

Response Letter

~~~~~

## Reviewer #1 (Remarks to the Author):

*The work presents a great experimental part that shows evidences of plasmonic anisotropy, however the points below need elucidation.*

**Reply:** We thank the reviewer for his/her careful reading and evaluation of our work. We have addressed all concerns in the revised version of the manuscript by displaying additional data and providing a more refined analysis where appropriate.

*1. The Raman analysis, used to show the crystalline anisotropy, is only superficially discussed. As anisotropy is an important feature for the work, one should further the explanation of the Raman spectra and clearly indicate, if possible, citing proper literature, which information confirms the anisotropy.*

**Reply:** We are grateful for the professional advice. 2M-WS2 possesses a low-symmetry monoclinic structure, resulting in anisotropic phonon vibrations. To reveal this anisotropy, we performed polarization-dependent Raman scattering measurements. For the measurement, the incident light was polarized along the *c*-axis of the crystal before rotating the sample, and a linear polarizer was placed in front of the detector to select the Raman signals parallel to the incident light polarization. Figure R1 presents the Raman spectra measured in unpolarized and parallel configurations. Seven Raman modes can be resolved, and the intensity of some Raman modes is sensitive to the light polarization as the sample rotates from 0° to 180°. In contrast to 2H-WS2, which has only two first-order Raman-active modes due to its trigonal prismatic crystal structure, 2M-WS2 exhibits more vibrational modes due to the reduction of the crystal structure symmetry. According to the literature, bulk 2M-WS2 belongs to the C2/*m* space group [*Adv. Mater.* 31, 1901942 (2019)], and

Raman active phonon modes involve  $A_g$  and  $B_g$  [*Thesis*, Fazel Baniasadi, (2021); *Adv. Mater.* 30, 1706771 (2018)]. Among them, the lowest frequency phonon mode  $B_g$  is an in-plane vibration mode with the polarization-dependent intensity expressed as:

$$I(B_g) \propto \sin^2 2\theta$$

where  $\theta$  is the angle between the light polarization direction and one of the sample principal axes. As shown in Fig. R1c, the angle-dependent intensity of the  $B_g$  mode is consistent with the Raman selection rule. In summary, the polarization-dependent Raman modes directly reflect the in-plane anisotropy of the 2M- $WS_2$  crystal.

**Fig. R1 Raman spectra of a bulk 2M- $WS_2$  crystal measured using a Horiba HR LabRam system with a laser wavelength of 532 nm. a** Raman spectrum of a bulk 2M- $WS_2$  crystal. **b** Polarized Raman spectra of the bulk 2M- $WS_2$  crystal. In the measurement process, a parallel polarization configuration for the polarizer and analyzer was adopted, and we rotated the sample from 0° to 180°, with the incident light polarized along the  $c$ -axis defined as 0°. **c** Angle-dependent intensity of the  $B_g$  mode.

Following the reviewer’s suggestion, in the revised Supplementary Information, we improved the Supplementary Fig. 1 as follows:

**Supplementary Fig. 1 | Raman spectra of a bulk 2M-WS2 crystal measured using a Horiba HR LabRam system with a laser wavelength of 532 nm. a** Raman spectrum of a bulk 2M-WS2 crystal, where seven Raman active phonons belonging to Ag and Bg modes are observed1-3. **b** Polarized Raman spectra of the bulk 2M-WS2 crystal. In the measurement process, a parallel polarization configuration for the polarizer and analyzer was adopted, and we rotated the sample from 0° to 180°, with the incident light polarized along the *c*-axis defined as 0°. **c** Angle-dependent intensity of the Bg mode, this result is consistent with the corresponding Raman selection rule  $I(B_g) \propto \sin^2 2\theta$ , as indicated in the reference1.

2. In the line 146 of the “Anisotropic infrared plasmons in 2M-WS2 disk arrays” section, the small difference between plasmons resonant frequencies of the two

*largest disks in the far-IR is explained by the retardation effect that is important when the 2D conductivity becomes comparable to the light velocity in Gaussian units. In this case, the plasmon dispersion is governed by the nanostructures' geometry and the dielectric environment according to the presented discussion.*

*This part rises some questions:*

*2.1 Which are the 2D conductivity values for those disks that allow assuming the retardation effect? Are these values different from those obtained for the fits in Fig. 1 for the bare crystal?*

**Reply:** We thank the reviewer for raising this question. The 2D conductivity mentioned in lines 151-156 in the manuscript refers to the intrinsic film conductivity. According to the literature, when the 2D conductivity becomes comparable to the light velocity in Gaussian units, electrodynamic effects rather than electrostatic govern 2D plasma dynamics. Specifically, retardation effect must be considered when the conductivity  $\sigma > c/2\pi = 5.3 \text{ mS}$  [*Sov. Phys. JETP* 68, 1150 (1989); *Phys. Rev. Lett.* 114, 106805 (2015)]. Fig. R2 illustrates the comparison between the calculated optical conductivity threshold and the real part of the intraband conductivity, which dominates charge relaxation. The estimated minimal conductivity threshold is basically consistent with our experimental results. Plasmons along the *b*-axis have larger frequency range in the retardation regime than those along the *c*-axis. Note that this is only a rough estimate, and the retardation effect could not cause an abrupt consequence. We can only determine that the retardation effect must be considered (or could be ignored) when the 2D conductivity is greater (or far less) than this estimated value. We emphasize that retardation in this work is not an assumption but a fact. In contrast to the blueshift of plasmons with increasing structural spacing under electrostatic conditions, plasmons in retardation regime exhibit a distinctive redshift with increasing structure spacing, as shown in Fig. R3. Meanwhile, the linear dispersion along the *b*-axis is a typical characteristic of the retardation effect [*ACS Nano* 5, 2535 (2011)], which sharply contrasts with the  $q^{1/2}$  dispersion for the case where retardation is negligible. Furthermore, in the electrostatic regime, the scaling

law of the plasmon frequency versus Drude weight is given by  $\omega \propto \sqrt{D}$ . Although with more than two times of the Drude weight along the  $b$ -axis compared to that along the  $c$ -axis, the experimentally observed frequency ratio between the plasmons along the two principal axes becomes smaller and smaller as the frequency decreases. This behavior is inconsistent with the aforementioned electrostatic condition. The small difference aligns with the condition for plasmons in retardation regime, where plasmon dispersions are primarily governed by the geometry of the nanostructures and the dielectric environments. And the experimentally observed behavior is also in accordance with the finding that plasmon dispersions coincide with each other for different metals in retardation regime [ACS Nano 5, 2535 (2011)].

**Fig. R2** Real part of the intraband (Drude) conductivity versus wavenumber, with the shaded area indicating the retardation regime.

**Fig. R3** Plasmon spectra of two disk arrays with the same disk diameter but different

spacings.

2.2 *If the retardation effect is relevant for the 8 and 6 micrometer-sized diameter disks, why does it not cause a similar influence on the smaller disks of nanometer-sized diameters?*

**Reply:** We thank the reviewer for this comment. For 2D plasmon in the non-retarded regime, the plasmon dispersion scales as  $\omega \propto \sqrt{q}$ , where  $q$  is the wavevector of the plasmon. For a fixed material, if  $q$  is small enough (corresponding to the localized plasmon in a large structure), resulting in the group velocity of the plasmon  $v_g = \partial\omega/\partial q \propto 1/\sqrt{q}$  larger than that of light, the corresponding plasmons are severely affected by the retardation effect [*J. Phys.: Condens. Matter* 33, 185302 (2021); *Phys. Rev. B* 92, 045434 (2015)].

2.3 *As the dielectric environment must be considered, how was the substrate computed in the modelling?*

**Reply:** We thank the reviewer for the question. According to the literature [*ACS Nano* 5, 2535 (2011)], the polarizability of an oblate spheroid in vacuum, considering dynamic depolarization and radiation damping, can be expressed as:

$$\alpha(\omega) = \frac{4\pi}{3} a^2 b \frac{\omega_p^2}{\left[ L\omega_p^2 - \omega^2 \left( 1 + \frac{\omega_p^2}{3c^2} ab \right) \right] - i\omega \left( \Gamma + \frac{2\omega_p^2 \omega^2}{9c^3} a^2 b \right)}$$

where  $a$  and  $b$  are the long and short axes of the spheroid,  $L$  represents the geometric depolarization factor,  $\omega_p$  is the bulk plasmon frequency of the material, and  $\Gamma$  is the scattering width for the intraband transition. For a disk with a diameter  $d$ , the parameter  $a$  in the formula is  $d/2$ , and  $b$  is the thickness of the disk. The plasmon is manifested by the resonance in the polarizability, therefore its frequency can be obtained by setting the real part of the polarizability denominator to zero. The result is particularly simple in the large disk limit (i.e., large  $a$ ):

$$\omega = \sqrt{\frac{6L}{b}} c \frac{1}{\sqrt{d}}$$

If the disk is sandwiched by two dielectrics with dielectric constants  $\epsilon_1$  and  $\epsilon_2$  ( $\epsilon_1 > \epsilon_2$ ), the light velocity  $c$  in the formula should be divided by a factor  $\sqrt{\epsilon_1}$  [EPL 99, 27006 (2012)]. As a result, the plasmon frequency is expressed as:

$$\omega \propto \frac{1}{\sqrt{\epsilon_1} d}$$

Therefore, we conclude that plasmon dispersions are primarily governed by the geometry of the nanostructures and the dielectric environments in the retardation regime for large structures.

*2.4 A more technical aspect: what is the spacing between disks in each patterning?*

**Reply:** The spacing between the disks is 0.65 times their diameter. We have added this parameter in the revised manuscript in lines 176-177.

*3. In the fit modelling to the bare crystals, it was used 1 term for the Drude response of free charges and 6 terms accounting for interband transitions. However, to extract the plasmons parameters from the disk-patterned crystals for a same spectral range, the fit modelling considers 3 three terms: 1 for the plasmon and 2 for interband transitions. Could the authors explain the reason the amount of interband components is different in each case? Based on the interband transitions fundamentals that is determined by the material electronic configuration, is it expected to vary due to the patterning in comparison to the bare crystal?*

**Reply:** Thanks for raising this concern. As the reviewer pointed out, the number of interband transitions is independent of the patterning. Due to the different measurement ranges for the exfoliated film and the disk arrays, we fitted the film conductivity and the plasmon spectra with different numbers of the interband transitions. For the spectra used to extract the film conductivity, the measurement range is about 600-20600  $\text{cm}^{-1}$  as shown in Supplementary Fig. 2. Within this range, six interband transitions exist. Including the visible light spectra range makes the fitted conductivity, especially for the high-frequency range of the near-infrared, more

accurate. However, for the plasmon spectra fitting procedure, since the highest plasmon frequency of the disks is lower than  $6500\text{ cm}^{-1}$ , we only measured the frequency range of  $600\text{-}12000\text{ cm}^{-1}$  ( $100\text{-}12000\text{ cm}^{-1}$  for the  $8\text{ }\mu\text{m}$  and  $6\text{ }\mu\text{m}$  disk arrays), in which range 3 relatively complete interbands are involved. Given that the spectra of the structures are dominated by response of the plasmons, and some of the interband transitions become indistinguishable after patterning, it is reasonable to fit the spectrum using one plasmon resonance and two interband transitions and get the plasmon frequency and linewidth, which are the key parameters we try to obtain.

We realize that the different measurement ranges of the film presented in the manuscript and the Supplementary Information may confuse readers. In the revised manuscript, in lines 116-119, we modified the sentence as “The measured extinction spectra were fitted using the optical conductivities of Drude response and interband absorptions, as shown in Supplementary Fig. 2 (see the details in Methods and the Supplementary Note 1).” to avoid potential misunderstanding.

4. *A question related to the previous, it is used the fit extracted conductivity from the unpattern crystal, which has the 6 interband components, as input to calculate the dispersion from the loss function. But the plasmon experimental dispersion is explained (fitted) by the modelling with 3 interband components. These approaches seem to have discrepancies.*

**Reply:** We thank the reviewer for the comment. As mentioned in the last question, to obtain the accurate film conductivity, we measured the spectra spanning from mid-infrared to visible light frequency range and fit the film conductivity with six interband transitions. The extracted film conductivity was then employed to calculate the loss function. For the patterned structures, infrared plasmon dominates the spectra with the highest frequency around  $6500\text{ cm}^{-1}$ . Therefore, measurements extending to the visible light frequency range will not cause discernable influence on the plasmon frequency fitting. Consequently, we only measured mid- to near-infrared spectra and fitted with one plasmon resonance and two interbands to extract the plasmon frequency and the linewidth. It should be noted that the plasmon frequency is what we

need to map out the plasmon dispersion and compare with that derived from the loss function.

5. *It is said that the loss function calculation considers the retardation effect, however there is no discussion on why it is needed. What would change in the dispersion if retardation was not taken into account?*

**Reply:** We thank the reviewer for this nice comment. Both the linear dispersion along the  $b$ -axis and small frequency differences between the two axes for plasmons of the relatively large disks reveal the existence of the retardation effect [*ACS nano* 5, 2535 (2011)]. Considering retardation leads to a dispersion redshift, as shown in the Fig. R4. This behavior has also been demonstrated in graphene plasmons with retardation effect [*EPL* 99, 27006 (2012); *Phys. Rev. B* 100, 155432 (2019)]. Despite the seemingly tiny difference for the calculated loss functions with and without retardation, accounting for the retardation effect in the loss function is the foundation for subsequent discussions of the long-range dipolar coupling.

**Fig. R4** The difference in the maxima of the loss function between the situation with and without retardation.

6. *On the isofrequency contours of Fig. 4.*

6.1 Could the authors describe the fitting modelling for skew ribbons? Which were the Drude weights and how many interband components were used in each case? Is this approach consistent with the modelling of the disks' cases?

**Reply:** We thank the reviewer for raising this question. For the skew ribbons, we measured and fitted the spectra when the electric field of the polarized infrared light was perpendicular to the ribbons, thus the Drude response was not included in the fitting procedures. All the fitting procedures for the plasmon spectra are the same. As described in the supplementary information:

To extract the parameters from the plasmon spectra in Fig. 2a-b in the main paper, the following model for the effective optical conductivity  $\sigma(\omega)$ , which also takes into account interband transitions, is employed:

$$\sigma(\omega) = i \frac{f}{\pi} \frac{\omega W}{(\omega^2 - \omega_p^2 + i\omega\Gamma_p)} + i \frac{f}{\pi} \sum_N \frac{\omega S_{inter,N}}{(\omega^2 - \omega_{inter,N}^2 + i\omega\Gamma_{inter,N})} \quad (S2)$$

where  $\omega_p$  and  $\Gamma_p$  ( $\omega_{inter,N}$  and  $\Gamma_{inter,N}$ ,  $N=1, 2$ ) are the frequency and linewidth of the plasmon (interband transition),  $W$  and  $S_{inter,N}$  are the spectral weights of plasmon and interband transitions, respectively, and  $f$  is the filling factor of the disk array.

To extract the plasmon frequencies of the skew ribbons, similar to the case of the disks for which the measurement range is about 800-12000  $\text{cm}^{-1}$ , two effective interbands were taken into account. For some of the spectra of the skew ribbons, only the range of 800-8000  $\text{cm}^{-1}$  was measured owing to the relatively low plasmon frequency. Given the measurement range as well as the plasmon dominance in the spectra and some of the interbands becoming indistinguishable after patterning, one Lorentz peak for plasmon and 1-2 effective interbands were used to extract the plasmon frequency.

In the revised Supplementary information, in lines 50-57, we described the extraction of the plasmon frequency of the skew ribbons as:

To extract the plasmon frequencies of the skew ribbons, similar to the case of the disks for which the measurement range is about 800-12000  $\text{cm}^{-1}$ , two effective interbands were taken into account. For some of the spectra of the skew ribbons, only

the range of 800-8000  $\text{cm}^{-1}$  was measured owing to the relatively low plasmon frequency. Given the measurement range as well as the plasmon dominance in the spectra and some of the interband becoming indistinguishable after patterning, in addition to the Lorentz peak for plasmon, 1-2 effective interband peaks were sufficient in the fitting to extract the plasmon frequency.

6.2 *How was  $q$  determined? In the disks' case it comes from the relation in line 187, but the one for ribbons is not explicitly informed in the text.*

**Reply:** We thank the reviewer for pointing out our omission of this definition. We have added the expression of  $q$  for ribbons in lines 249-251 in the manuscript as follows:

Consequently, the dispersions of the plasmons with wave vector  $q$  ( $q = \pi/L$ ,  $L$  is the ribbon width) along different directions are plotted, as shown in Fig. 4a.

6.3 *The theoretical dispersion curves in Fig. 4b are obtained by fitting the loss function maxima to the experimental data. This procedure corresponds to determining the material conductivity that is a material constant, only changeable under extreme states (high magnetic fields, temperature, and pressure, for instance), and was already evaluated from the fits in Fig. 3a. It is not clear such a change in the conductivity from the disk to the ribbon patterned  $\text{WS}_2$ . This point is very critical.*

**Reply:** Thanks for this comment. For the anisotropic film, the loss function is a function of the frequency  $\omega$  and  $\mathbf{q}$ , with  $\mathbf{q}$  in the 2-dimensional wave vector space. So it's a comprehensive multi-variable function. The referee is right. The loss function is solely determined by the anisotropic optical conductivity of the intrinsic film itself, regardless the structure shape patterned to host the localized plasmon in the experiment. However, since the plasmon dispersion is different for  $\mathbf{q}$  along different directions, the comprehensive loss function can reflect this. For instance, Fig. 3a exhibits the dispersions derived from the loss function along two high symmetry

directions ( $0^\circ$  and  $90^\circ$ ). In the disk case, when the electric field of the polarized light is along directions other than the principal axes, the measured spectrum is the linear combination of two plasmon modes along the two principal axes. As a result, dispersions along an arbitrary direction cannot be obtained through measuring spectra of disk arrays. Nevertheless, as an isotropic structure, disk is convenient for characterizing the anisotropy of the plasmons as well as their dispersions based on the intrinsic properties of the material. Now, in Fig. 4a, to determine the isofrequency contours, we show the dispersions with  $\mathbf{q}$  along other directions. To achieve this in the experiment, dispersions along different directions between the principal axes were obtained through measuring plasmons of skew ribbons. In the meantime, the effective conductivity along each specific direction was adopted in the corresponding loss function calculation. The effective optical conductivity for a direction with the skew angle of  $\theta$  relative to the  $b$ -axis could be expressed as:  $\sigma_L = \sigma_b \cos^2 \theta + \sigma_c \sin^2 \theta$  ( $\sigma_b$  and  $\sigma_c$  are the sheet conductivities along the  $b$ - and  $c$ -axis, respectively). Thus, measured and calculated dispersions based on the loss function along different directions can be plotted for the skew ribbon case.

In the revised manuscript, we disentangled the motivation for fabricating disks and ribbons, as described in lines 143-145 and lines 238-243:

This geometrically isotropic structure enables us to characterize the plasmons by far-field spectroscopy and reveal the anisotropy of plasmons resulted from the intrinsic properties of the material.

To experimentally determine the plasmon isofrequency contours of the 2M-WS2 in the two-dimensional momentum space through far-field spectroscopy, it's necessary to obtain plasmon frequencies with wave vector  $\mathbf{q}$  along various directions as well. Microstructure arrays in the ribbon form are specifically suitable to meet this purpose, rather than disks with plasmons only along the principal axes measurable in experiment.

6.4 *Which was the spacing in between the ribbons for each angle? Could the authors comment on how the spacing would change the dipole coupling, which is*

discussed in supplementary information note 5, and that implied in the 0.43 scaling factor to adjust the isofrequencies contours in Fig. 4?

**Reply:** Thanks for this nice suggestion. We fabricated the ribbon array with a spacing of  $\sim 1.3$  times the ribbon width. A larger spacing will result in a larger redshift in the retardation regime [*Proc. Natl. Acad. Sci. U.S.A.* 106, 19227 (2009); *J. Chem. Phys.* 120, 10871-10875 (2004)]. This behavior is evident in Fig. R3, which shows plasmon redshift along the  $b$ -axis when increasing the spacing. Furthermore, by fabricating a CVD-grown large-area PtTe2 film into two series of ribbon arrays with filling factor of 50% and 35%, we can also confirm that dispersions should be scaled by a factor for structures with different spacings in the retardation regime, as shown in Fig. R5.

In the revised Supplementary information, we added the spacing of the ribbons in line 195 as “The spacing is  $\sim 1.3$  times the ribbon width.”

**Fig. R5** Plasmon dispersions for two series of PtTe2 ribbons with different filling factors ( $f$ =ribbon width/period).

6.5 The shown canalization in Fig. 4b is not reproduced by the theoretically predicted ones in supplementary Fig. 4. Could the reasons for that be presented?

**Reply:** We thank the reviewer for the comment. The difference between Fig. 4b in the main paper and supplementary Fig. 6b arises from neglecting dipole-dipole coupling in the loss function calculation for the latter figure. The dispersions displayed by the maxima of the loss functions in supplementary Fig. 6a are only governed by the intrinsic conductivity of 2M-WS2 films, and we plotted the corresponding isofrequency contours in supplementary Fig. 6b. However, after patterning, the dipole-dipole coupling from each structure in the retardation regime will typically result in the plasmons redshift [*Proc. Natl. Acad. Sci. U.S.A.* 106, 19227 (2009); *Materials Science and Engineering: B* 149, 251 (2008)]. This behavior is also evidenced in our experiment, as shown in the comparison of supplementary Fig. 6a and Fig. 4a. Given the aforementioned dipole-dipole coupling induced plasmon dispersion redshift after patterning when the conductivity along *b*-axis is included in the total conductivity, a larger momentum *q* is needed for the same plasmon frequency in experiment. Thus, the isofrequency contours seem to be elongated, especially along the *b*-axis in Fig. 4b in the main paper.

This discrepancy has been explained in Supplementary Note 5 (lines 175-181) as follows:

Specifically, given the aforementioned dipole-dipole coupling induced plasmon dispersion redshift after patterning when the conductivity along *b*-axis is included in the total conductivity, a larger momentum *q* is needed for the same plasmon frequency in experiment. Thus, the isofrequency contours seem to be elongated, especially along the *b*-axis in Fig. 4b in the main paper, compared to the loss function calculated one shown in Supplementary Fig. 6b (considering only the intrinsic conductivity of 2M-WS2 films).

## **Reviewer #2 (Remarks to the Author):**

*There has been substantial interest in creating a toolbox of materials which could be valuable in developing planar structures to control IR radiation for sensing and advanced photonics. 2D materials with natural anisotropies in the IR are ideal candidates in this space. The paper explores 2M-WS2 for this application. This work in many ways simply builds on earlier work on WTe2 by the same group. (Ref. Nat Commun. 11, 1158 (2020) and Light Sci. Appl. 12, 193 (2023)). The study shows that this materials system is anisotropic and can support plasmon polaritons in the 2-3micron range (3000cm-1 - 5000cm-1). While these observations are noteworthy, they are not substantially different than earlier work. One interesting new feature is the ability to tune the system using electrically driven ion-intercalation. This strategy for active control is both novel and interesting. Several questions need to be addressed.*

**Reply:** We thank the reviewer for careful reading and for pointing out the novelty of tuning plasmons through electrically driven ion-intercalation. In this study, we demonstrate the high in-situ tunability of the absorption of the 2M-WS2 films, and the frequency and amplitude of the plasmons along the two principal axes. More importantly, the ion intercalation method used in this work is reversible. Our demonstration sheds new light on the feasibility of tunable plasmonic materials with a thickness on the order of tens of nanometers, beyond electrical gate-tunable few-layer materials. We also highlight the superiority of layered plasmonic materials in terms of tunability compared to noble metals, which remain challenging to tune even for films with only several nanometers.

In addition, we want to emphasize the differences between this work and our earlier works based on semimetal WTe2. Firstly, anisotropic plasmons in 2M-WS2 operate from far- to mid-infrared regime at room temperature, whereas the anisotropic plasmons in WTe2 work in the frequency range of far-infrared at liquid helium temperature. Secondly, broadband plasmon canalization is demonstrated in 2M-WS2

in the mid-infrared regime, while terahertz hyperbolic plasmons are supported by thin films of WTe2.

*1. First there is a substantial difference in the resonance peaks in figure 2 between the simulation figure 2C and figures 2A&B for ~1micron diameter structures. No explanation is offered. This needs to be addressed.*

**Reply:** We thank the reviewer for pointing out the difference. The difference between the experimental results and the simulations can be mainly attributed to the dielectric environment. A substrate with a large dielectric constant tends to cause plasmons redshift. The samples are fabricated on diamond substrates with the thickness of about 1 mm. The significant contrast in thickness between the substrate and the sample with several tens of nanometers thick, effectively avoids interference fringes in the spectra. However, large simulation objects consume the memory of a computer. Therefore, we simulated a disk suspended in air to demonstrate the anisotropy of plasmons in 2M-WS2. The simulated frequency ratio of plasmons along *b*- and *c*-axis is about 1.36, which is basically consistent with the corresponding value of ~1.27 observed in the experiment. To avoid Wood's anomaly caused by periodic boundary conditions with an infinite array in the simulation, which is negligible in experiment with only thousands of structures, we simulated a single disk rather than a disk array. In summary, the different peak positions induced by the change in the dielectric environment and the influence of neglected dipole coupling contribute to the discrepancy between the experiment and simulation. Nevertheless, the most important features, such as the strong anisotropy, are well reproduced.

In addition to the statement we have mentioned in the Supplementary Information: “To avoid interference and save computer memory, the diamond substrate with a thickness of about 1 mm used in the experiment was not taken into account in the simulation.”, we have modified the sentences in lines 161-166 in the manuscript to highlight reproduction of the anisotropic feature of the plasmons in 2M-WS2 as follows:

Figure 2c displays the finite element simulated extinction spectra of a disk suspended

in air, based on the fitting parameters of the room temperature unpatterned film in Fig. 1. The simulated frequency ratio of plasmons along  $b$ - and  $c$ -axis is about 1.36, which is consistent with the corresponding experimental value of  $\sim 1.27$ . (see Supplementary Note 2 for the detailed information).

2. *The authors should focus more on the tunability aspect of this system, there are several interesting features in their data which they don't try to explain. For example, what drives the  $O\sim 1000\text{cm}^{-1}$  red shift observed in figure 5D? It's not as pronounced in figure 5E.*

**Reply:** We thank the reviewer for his/her insightful comment. Due to rapid decrease in intensity of the plasmon induced by large damping along the  $c$ -axis after intercalation, significant frequency redshifts were not observed in the original Fig. 5e. This behavior is probably resulted from the sample quality and seemingly inconsistent with the expected large tunability of the carrier density. To address this issue, we fabricated new samples and measured the infrared spectra, as illustrated in Fig. R6. With increasing intercalation duration, the plasmon along the  $c$ -axis redshifts from  $2200\text{ cm}^{-1}$  to a frequency below  $800\text{ cm}^{-1}$ , which is the lower measurable limit with  $\text{CaF}_2$  substrate. Such a substantial tunability (decrease to  $\sim 0.36$  times of the original frequency) agrees with the change in Drude weight extracted from the intercalation dependent extinction spectra along the  $c$ -axis of the 2M- $\text{WS}_2$  film, as depicted in Fig. 5g and Supplementary Fig. 10 (these figures are attached in this response letter on page 22 and page 23). The fitted Drude weight  $D$  decreased from  $9.59 \times 10^{12}\ \Omega^{-1}\text{s}^{-1}$  to  $9.06 \times 10^{11}\ \Omega^{-1}\text{s}^{-1}$  with increasing intercalation time. Therefore, by considering the relation between the plasmon frequency and the Drude weight for a 2D electric gas  $\omega \propto \sqrt{D}$ , the plasmon frequency decreases to 0.31 times of the original value in our measurement range after intercalation.

We take the suggestion from the referee and focus more on the tunability in the revised manuscript. We have analyzed the tunability of the plasmons along the  $c$ - and  $b$ -axis, quantifying the decrease in the Drude weight of the film, as shown in the

revised Fig. 5 and the corresponding description.

**Fig. R6** The evolution of the plasmon along the  $c$ -axis in a  $0.9 \mu\text{m}$  wide ribbon array under gate controlled deionized water.

3. *This work is of interest to the community but the authors should focus on the truly novel parts of their study. Restructuring the paper with increased focus in the viability of ion-gel based tunability would make this a stronger submission.*

**Reply:** We thank the reviewer for his/her nice suggestion, and for recognizing the interest in the topic and the novelty of our idea to study the ion-gel based tunability of plasmon. We have added additional data and conducted corresponding quantitative analyses to further elucidate the tunability of plasmons and the absorption of the 2M-WS2 films through ion intercalation. The new figures, including Fig. 5d-i, Supplementary Fig. 10, Supplementary Fig. 11, and Supplementary Fig. 13, are presented in this response letter on pages 22-25. These results strongly support the feasibility of ion intercalation-based tunability. Specifically, our findings demonstrate a reversible decrease in free carrier Drude weights by approximately one order of magnitude, leading to significant redshifts of the plasmon and notable modulation of the isofrequency contours, as illustrated in Fig. 5. Meanwhile, we have improved the abstract, introduction, and discussion sections to align with the more comprehensive

results related to the ion intercalation-based tunability.

In the revised manuscript, we modified the section “Reversibly tunable plasmons in 2M-WS2” as follows:

Two-dimensional materials provide a great platform for developing tunable photonic devices. To achieve dynamic tunability in photonic devices made of thin films, ion intercalation is more effective than regular electrostatic gating. Figure 5a depicts a schematic of gate-controlled ion intercalation through ion-gel. Figure 5b shows the extinction spectra of Li+ intercalated ribbons parallel to the *c*-axis, with the ribbon width of 0.6 μm and the electric field of the polarized infrared light along the *b*-axis. With increasing resistance, the plasmon redshifts and the strength decreases. Importantly, the tunability of the optical response and the resistance is reversible, as shown in Fig. 5c (see Supplementary Fig. 9 for more information). The highly tunable carrier density of 2M-WS2 also leads to the modulation of the film absorption. As shown in Supplementary Fig. 8b and Fig. 8c, a film with the source and drain along one of the principal axes is intercalated via Li+ ion-gel, and the extinction spectra exhibit a reduction of the Drude response as the resistance of the film increases. Due to the large Li+ induced damping along the *c*-axis which is manifested by the extinction spectra of the film along the *c*-axis, plasmon shifts cannot be detected before the plasmons fully diminish in strength.

Other media for the intercalation are also possible. We further tuned the carrier density of the 2M-WS2 through H+ intercalation from gate-controlled electrolyzed deionized water62. To avoid extra absorption, water droplets were blown off and samples were dried through nitrogen gas before measurements. As shown in Fig. 5d and 5e, the tunability of plasmons along the *b*- and *c*-axis is also demonstrated. The plasmons redshift and the strength decreases with increasing resistance. The redshift of the plasmon along the *b*-axis is ~17% of the original frequency (Fig. 5d), similar to the corresponding value of ~20% in Fig. 5b with Li+ ion-gel. The plasmon along the *c*-axis redshifts from 2200 cm-1 to a frequency below 800 cm-1, which is the lower measurable limit with CaF2 substrates. The amount of the frequency redshift is larger than 63% along the *c*-axis, manifesting significant tunability of ion intercalation.

Figure 5f presents the resistance-dependent frequency of the plasmons. The change of the plasmons with ion intercalation can be further confirmed through extinction spectra of the intercalated unpatterned 2M-WS2 film. As shown in Supplementary Fig. 10 and Supplementary Fig. 11, intercalation dependent extinction spectra of a film and the corresponding fitting results are presented. Drude weights decrease with increasing ion intercalation, while the peak positions and the strength of the interband transitions remain substantially fixed, simplifying the analysis of the intercalation effect on the plasmon.

The extracted Drude weight along the *b*-axis decreases linearly with that of the *c*-axis, as shown in Fig. 5g. The slope is about 2.6, which is consistent with the ratio of  $D_b/D_c=2.6$  extracted from Fig. 1d. With the Drude weight decreasing to 0.1 (0.15) times of the original along the *c*-axis (*b*-axis) after intercalation, the corresponding loss function changes from Fig. 3a to Fig. 5h. A significant redshift of the plasmon along the *c*-axis with the value of ~88% is in agreement with the experimental result in Fig. 5e. More importantly, with the Drude weight decreasing to about one order smaller than the original value and the interbands almost intact, the original canalization frequency range shifts to lower frequency. We take the lower canalization frequency range limit of 3000 cm-1 as an example to demonstrate the change of the calculated isofrequency contours. As shown in Fig. 5i, the elliptic isofrequency contour gradually enlarges and eventually changes to a flat shape as the intercalation progresses (see Supplementary Note 6 for detailed information). The extinction spectra after deintercalation and additional spectra of ion intercalated 2M-WS2 films ensuring the reproducibility of the tuning are displayed in Supplementary Fig. 9 and Fig. 13.

In the revised abstract in the manuscript, in lines 42-44, we modified the original sentence to emphasize the ion intercalation tunability as “Furthermore, the anisotropic plasmons and the corresponding isofrequency contours can be reversibly tuned via in-situ ion-intercalation.”.

In the revised introduction in the manuscript, in lines 72-74, we highlighted the

importance of the tunable photonic device as following:

Therefore, natural in-plane anisotropic plasmons with high in-situ tunability are strongly desirable in low-symmetry photonics applications2-5, 29-51.

We also added a sentence to describe the current achievement related to in-situ tuning, as expressed in lines 82-83:

Meanwhile, electrical tuning of plasmons in aforementioned films remains challenging.

In lines 97-99, the original sentence is revised as “More importantly, ion-intercalation renders anisotropic plasmons and the corresponding isofrequency contours reversibly tunable.”

In the revised discussion in the manuscript in lines 350-353, we modified the original sentences as “Furthermore, reversibly tunable spectral responses of the plasmons and the corresponding isofrequency contours via in situ ion-intercalation are demonstrated. Our study paves the way for exploring the electrically switchable unidirectional plasmonic devices.”

**Fig. 5 | Tunable plasmons in 2M-WS2 through ion-intercalation.** **a** An illustration of the infrared spectrum measurement of an ion intercalated sample. **b** The evolution of the plasmon spectrum in a 0.6 μm wide ribbon array with ribbons along *c*-axis under gate-controlled Li+ intercalation. Due to the strong absorption of the ion-gel below 1500 cm-1 and the other region with shade of gray in the figure, these frequency regimes are not displayed. **c** Reversible resistance-dependent plasmon frequency extracted from **b**. **d-e** The evolution of the plasmons in ribbon arrays along the *b*- and *c*-axis under gate controlled deionized water, respectively. **f** Resistance-dependent plasmon frequencies extracted from **d** and **e**. **g** Extracted Drude weights of a film along the *b*- and *c*-axis at different intercalation durations. They are proportional to each other. **h** Calculated loss function using the extracted film parameters after intercalation. **i** Calculated intercalation dependent isofrequency contours at the

frequency of  $3000\text{ cm}^{-1}$ .

**Supplementary Fig. 10** | a  $c$ -axis polarized extinction spectra of a 2M-WS2 film with ion intercalation-controlled resistances. The thickness of the film is about 40 nm.

**b-k** Fitting results of the spectra in **a**. **I** Comparison of spectra between the sample before the intercalation and after the deintercalation process.

**Supplementary Fig. 11 | a** Extinction spectra of a 2M-WS2 film along the *b*-axis with ion intercalation-controlled resistances, the thickness of the film is about 40 nm. **b-k** Extinction spectra fitting results of the film of 2M-WS2 along the *b*-axis. **l** Comparison of spectra between the sample before the intercalation and after the deintercalation process.

**Supplementary Fig. 13 | Repeatability of the ion intercalation through gate-controlled deionized water.** Extinction spectra of a 2M-WS2 film along the *b*- and *c*-axis with ion intercalation and deintercalation. The thickness of the film is about 45 nm.

*I believe this paper would be worthy of publication with substantial revision.*

**Reply:** We have improved our manuscript comprehensively. We appreciate the very constructive and knowledgeable comments from the reviewer.

~~~~~

Reviewer #3 (Remarks to the Author):

1. *This work utilized far-field spectroscopy on nano-patterned devices to demonstrate an anisotropic plasmon in 2M-WS₂ thin films.*
2. *The authors systematically studied the isofrequency contours of the TMD material and demonstrated an elliptical plasmon polariton dispersion with broadband canalization.*
3. *The authors further demonstrated the tunability of the anisotropic plasmonic response using ion gel gating.*
4. *Prior work mainly focuses on exciton-polariton in TMD materials and this work is the first to demonstrate tunable anisotropic plasmon polariton induced by the Drude and interband response.*
5. *Compared to prior works from their group, this work demonstrated the tuning of plasmon polariton through ion gel in the mid-infrared range. This work also demonstrated broadband canalization of plasmon that enables unidirectional propagation.*

Reply: We thank the reviewer for his/her careful reading and evaluation of our work.

The authors could comment on whether there are other ways to tune the system, such as optically induced hyperbolicity [Sternbach et al., Science 371, 617–620 (2021)].

Reply: We thank the reviewer for providing this valuable reference. While for intrinsic 2M-WS₂, it may be challenging to tune through laser pumping due its high carrier density ($\sim 3.2 \times 10^{21} \text{ cm}^{-3}$ at room temperature), after intercalation, the carrier density can be significantly reduced. A prospective hyperbolic range, where the imaginary part of the optical conductivity satisfies $\sigma_b''/\sigma_c'' < 0$, undergoes shift, as shown in Fig. R7. Considering optical pumping with the highest attainable photo-carrier density of $\sim 1.8 \times 10^{20} \text{ cm}^{-3}$ [*Phys. Rev. Lett.* 126, 227402 (2021)], in combination with the ion intercalation, tunable hyperbolicity is possible as the Drude weight varies with pump fluence.

Additionally, transport experiments have verified that 2M-WS₂ can be tuned by applying pressure and changing temperature [*npj Quantum Materials* 4, 50 (2019); *Adv. Mater.* 31, 1901942 (2019)]. Although the carrier density only decreases to

one-third of that at room temperature with the temperature decreases to 10 K. After intercalation, with a lower carrier density, the temperature effect may be more significant. Therefore, applying high pressure or changing temperature combined with the ion intercalation is also potentially capable to tune this system.

Fig. R7 The frequency dependence of the imaginary part of optical conductivity ratio between two axes with decreasing Drude weights. The Drude weights D_b and D_c are extracted from Fig. 1d in the main paper, and the coefficients come from the fitting results of Supplementary Fig. 11 and Fig. 10, which indicates one-to-one linear decrease of the Drude weights. The light blue shaded areas indicate optical conductivity ratio with $\text{Im}(\sigma_b) > 0, \text{Im}(\sigma_c) > 0$ and $\text{Im}(\sigma_b)/\text{Im}(\sigma_c) > 3$. The light gray shaded areas represent the range where $\text{Im}(\sigma_b)/\text{Im}(\sigma_c) < 0$.

This work should be published with minor revisions.

Reply: We sincerely thank the reviewer for positive remarks and for recommending publication of our work.

REVIEWERS' COMMENTS

Reviewer #1 (Remarks to the Author):

The authors have satisfactorily responded to the raised points needing clarification. My recommendation is for publication.

Reviewer #2 (Remarks to the Author):

The authors have adequately satisfied my concerns. I believe the manuscript is ready for publication.